# Multi-Head Adapter Routing
# for Cross-Task Generalization

**Lucas Caccia**■■■   **Edoardo Ponti**■   **Zhan Su**■   **Matheus Pereira**■

**Nicolas Le Roux**■■■■   **Alessandro Sordoni**■■■

■Microsoft Research, ■McGill University, ■MILA,
■University of Edinburgh, ■Université de Montréal, ■University of Copenhagen
`lucas.page-caccia@mail.mcgill.ca,alsordon@microsoft.com`

## Abstract

Parameter-efficient fine-tuning (PEFT) for cross-task generalization consists in pre-training adapters on a multi-task training set before few-shot adaptation to test tasks. Polytropon [Ponti et al., 2023] (`Poly`) jointly learns an inventory of adapters and a *routing* function that selects a (variable-size) subset of adapters for each task during both pre-training and few-shot adaptation. In this paper, we investigate the role that adapter routing plays in its success and design new variants based on our findings. First, we build on the intuition that finer-grained routing provides more expressivity. Hence, we propose `MHR` (Multi-Head Routing), which combines *subsets* of adapter parameters and outperforms `Poly` under a comparable parameter budget; by only fine-tuning the routing function and not the adapters (`MHR-z`), we achieve competitive performance with extreme parameter efficiency. Second, we find that `Poly/MHR` performance is a result of better multi-task optimization, rather than modular inductive biases that facilitate adapter recombination and local adaptation, as previously hypothesized. In fact, we find that `MHR` exhibits high gradient alignment between training tasks. We find that routing is most beneficial during multi-task pre-training rather than during few-shot adaptation and propose `MHR-`$\mu$, which discards routing and fine-tunes the average of the pre-trained adapters on each downstream tasks. This establishes `MHR-`$\mu$ as an effective method for single-adapter fine-tuning. We also show that `MHR-`$\mu$ can be used as an effective zero-shot transfer method by training the average of the pre-trained adapters for a few additional steps on the multi-task training set: this yields gains up to 3% on absolute accuracy w.r.t. the baselines.

## 1 Introduction

The ability to train effective models with a relatively small number of training data is of paramount importance due to the paucity of annotated examples for most tasks. One effective few-shot learning approach is to leverage large models pre-trained on a vast amount of unlabelled data and fine-tune them on the few examples available for each downstream task. To reduce the memory cost of duplicating the entire array of parameters for each downstream task, recent approaches resort to parameter-efficient fine-tuning (PEFT) methods, such as LoRA [Hu et al., 2022], SFT [Ansell et al., 2022], or (IA)³ [Liu et al., 2022]. These only fine-tune adapters while leaving the pre-trained model 'frozen'.

Nevertheless, it remains unclear how to best exploit a set of *training* tasks to better generalize to a set of unseen *test* tasks in a sample-efficient fashion, based on just a few examples. One straightforward

37th Conference on Neural Information Processing Systems (NeurIPS 2023).

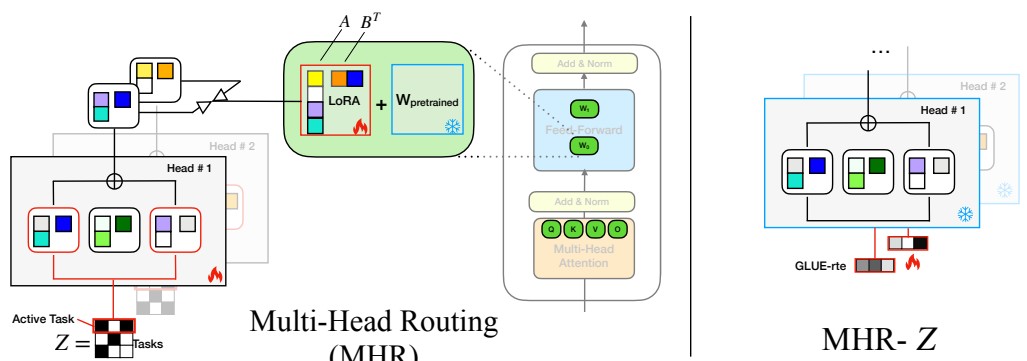

Figure 1: *Left:* A LoRA adapter with weight $AB^\top$ is trained on top of a frozen, pre-trained linear layer $W$. Our method `MHR` partitions the $A, B$ parameter indexes into $h$ subsets (or *heads*). For each subset, a separate routing function selects the active modules for the current task among $m$ copies with different parameter values, and combines them via averaging to form a task-specific head. The heads are then concatenated to form the LoRA adapter. Using multiple heads allows for more fine-grained mixing of task parameters with a negligible increase in overall parameter count. *Right:* During few-shot adaptation, one can fine-tune only the multi-head routing parameters (`MHR-z`), keeping the modules frozen, resulting in highly parameter-efficient adaptation.

solution is to perform multi-task pre-training, i.e. first train the large model on the union of the examples from the training tasks, then fine-tune the obtained model to the test task [Liu et al., 2022, Ye et al., 2021]. However, this solution does not take into account that test tasks may require solving different combinations of sub-problems compared to training tasks [Vu et al., 2020], thus failing to achieve compositional generalization [Rosenbaum et al., 2019, Ponti, 2021]. Moreover, specializing the model towards different tasks during training may result in negative transfer, due to their corresponding gradients being misaligned [Wang et al., 2021].

Several PEFT approaches have been proposed to enable better cross-task generalization by training adapters (or soft prompts) on each task independently [Pfeiffer et al., 2021, Vu et al., 2021, Asai et al., 2022, Chronopoulou et al., 2023]. Given a new test task, parameters from similar training tasks are aggregated, which enables transfer. While solely having task-specific parameters is an effective strategy to mitigate interference across training tasks, it also inhibits any positive transfer within the same task pool. Polytropon (`Poly`) was recently proposed by Ponti et al. [2023] to address these issues: the model assumes that task-specific adapters are learned combinations of a reusable inventory of basis adapters or *modules*. In practice, each module is implemented as a LoRA [Hu et al., 2022] adapter, which modifies a large pre-trained model, such as T5 [Raffel et al., 2020]. During both multi-task pre-training and few-shot adaptation, `Poly` learns both the inventory of adapters and a (continuously relaxed) binary task–module routing matrix, which determines which module is active for each task. While `Poly` shows promising results, several questions remain unanswered: 1) Does the expressivity of the routing function matter? 2) Why do routing-based PEFT methods yield superior performance? 3) Is routing useful during both multi-task pre-training and few-shot adaptation?

To answer the first question, we propose a new routing function, `MHR`, that mixes adapters at a more granular level. Differently from `Poly`, where routing decisions are made for each adapter as a whole, in `MHR` we linearly combine subsets of the adapter dimensions (i.e. heads), each with different combination coefficients. We evaluate `MHR` and a series of competitive baselines for few-shot task adaptation on the T0 task suite [Sanh et al., 2022] and Super-Natural Instructions [SuperNI; Wang et al., 2022a]. Based on our results, we report that `MHR` outperforms `Poly` and single adapter baselines. Additionally, we show that, thanks to the increased expressivity of the routing function, it becomes possible to fine-tune only the parameters of the routing function (and not the adapters) during few-shot adaptation: the resulting method, `MHR-z`, yields competitive performance while requiring orders of magnitude fewer parameters.

Regarding the second and third questions, we uncover that optimization during multitask pretraining plays a key role in explaining the downstream performance of routing-based PEFT approaches.

Specifically, we find that `MHR` exhibits a higher cosine similarity between gradients from different tasks than `Poly` and single-adapter multi-task training. Hence, routing enables more knowledge transfer and less interference across tasks during multi-task pre-training. This finding led us to investigate whether routing is useful also during few-shot adaptation. It has been hypothesized [Ponti et al., 2023] that one of the reasons behind `Poly`'s performance resides in the inductive bias of the modular architecture, which allows test tasks to recombine and locally adapt the most relevant modules. To test this hypothesis, we propose `MHR`-$\mu$, where the routing function is discarded and all available adapter parameters are averaged before few-shot adaptation. We find that `MHR`-$\mu$ can recover the performance of `MHR`, hinting that `Poly`/`MHR` gains are only a result of better multi-task optimization. Finally, we show that `MHR`-$\mu$ can also be used as an effective zero-shot transfer method by training the average of the pre-trained adapters for a few additional steps on the multi-task training set. This yields gains up to 3% on absolute accuracy w.r.t. to strong baselines such as T0-11B.

## 2  Background

In cross-task generalization, we are given a set of tasks $\mathcal{T} = \{\mathcal{T}_1, .., \mathcal{T}_{|\mathcal{T}|}\}$, with each task $\mathcal{T}_i$ dataset containing a set of samples $\mathcal{D}_i = \{(x_1, y_1), ..., (x_n, y_n)\}$. The set of all tasks is partitioned into training and test tasks, $\mathcal{T} = \mathcal{T}_{train} \cup \mathcal{T}_{eval}$, and the objective is to leverage data in $\mathcal{T}_{train}$ and transfer knowledge to facilitate learning of the test tasks $\mathcal{T}_{eval}$. For all the methods discussed, learning takes place in two phases, excluding the original unsupervised pre-training of the language model backbone on a separate corpus. The first phase consists of multi-task pre-training, in which either an adapter, such as LoRA or (IA)$^3$, or the full backbone is trained on the set of training tasks $\mathcal{T}_{train}$. The second phase consists in few-shot adaptation, where the learned adapters are fine-tuned independently on each test task in $\mathcal{T}_{eval}$. We follow the procedure from [Raffel et al., 2020] and formulate each task as a text-to-text problem, enabling standard maximum-likelihood training with teacher forcing [Bengio et al., 2015] and a cross-entropy loss.

### 2.1  Adapters: `LoRA` & `(IA)`$^3$

LoRA [Hu et al., 2022] and (IA)$^3$ [Liu et al., 2022] are two recently proposed adapter architectures that achieve competitive trade-offs between performance and parameter efficiency [Karimi Mahabadi et al., 2021, Liu et al., 2022]. For each linear transformation corresponding to the query ($q$), key ($k$), value ($v$) and output ($o$) of the self-attention layers, LoRA modifies the base model *parameters* as follows:

$$h^{q,k,v,o} = \boldsymbol{W}_0^{q,k,v,o}x + s \cdot \boldsymbol{A}^{q,k,v,o}(\boldsymbol{B}^{q,k,v,o})^\top x, \qquad \text{(LoRA)}$$

where $\boldsymbol{W}_0$ are the (frozen) weights of the pre-trained model (e.g. T5 [Raffel et al., 2020]). $\boldsymbol{A}, \boldsymbol{B} \in \mathbb{R}^{d \times r}$ are low-rank learnable parameters and $s \geq 1$ is a tunable scalar hyperparameter. (IA)$^3$, on the other hand, modifies key and value *representations* in self-attention element-wise, and also modifies the feed-forward MLP ($f$):

$$h^{k,v} = \boldsymbol{l}^{k,v} \odot (\boldsymbol{W}_0^{k,v}x); \; h^f = (\boldsymbol{l}^f \odot \gamma(\boldsymbol{W}_1^f x))\boldsymbol{W}_2^f, \qquad \text{((IA)}^3\text{)}$$

where $\boldsymbol{l}^{k,v,f} \in \mathbb{R}^d$ are learnable parameters , $\boldsymbol{W}_{1,2}^f$ the frozen parameters of the feed-forward layer in the backbone, and $\gamma$ a non-linearity. For clarity, we will drop the superscripts $q, k, v, o$ in the rest of the paper.

### 2.2  Polytropon: Adapter Routing

Typical adapter methods either fully share adapters across tasks or train individual adapters for each task. `Poly` addresses the multi-task problem by softly sharing adapter parameters across tasks. Each `Poly` layer contains 1) an inventory of adapter modules $\mathcal{M} = \{\phi_1, \ldots, \phi_m\}$ with $|\mathcal{M}| \ll |\mathcal{T}|$; 2) a routing function $r(\cdot)$ that chooses which subset of the modules to combine for each task.

Each module corresponds to a LoRA adapter, where $\phi_i$ are its associated parameters $\boldsymbol{A}^{(i)}, \boldsymbol{B}^{(i)} \in \mathbb{R}^{d \times r}$. $r(\cdot)$ is implemented as a task–module routing matrix $\boldsymbol{Z} \in \mathbb{R}^{|\mathcal{T}| \times |\mathcal{M}|}$. $z_\tau = \boldsymbol{Z}_{\tau,:} \in \mathbb{R}^{|\mathcal{M}|}$ is a routing vector of task $\mathcal{T}_\tau$, with cell $Z_{\tau,j}$ being the probability logits of using module $\phi_j$ for task $\mathcal{T}_\tau$ in the current layer. Differently from mixture-of-experts [Fedus et al., 2022], which perform token-level top-$k$ routing, $\boldsymbol{Z}$ converges to a binary matrix, defining a soft partition over modules.

| Method | Pre-Training | Fine-Tuning | Inference |
|---|---|---|---|
| Full FT | $d \times d$ | $d \times d$ | $d \times d$ |
| LoRA | $d \times 2r$ | $d \times 2r$ | $d \times 2r$ |
| Poly | $d \times 2r \times |\mathcal{M}| + |\mathcal{T}| \times |\mathcal{M}|$ | $d \times 2r \times |\mathcal{M}| + |\mathcal{M}|$ | $d \times 2r$ |
| Poly-$z$ | $d \times 2r \times |\mathcal{M}| + |\mathcal{T}| \times |\mathcal{M}|$ | $|\mathcal{M}|$ | $|\mathcal{M}|$ |
| MHR-$\mu$ | $d \times 2r \times |\mathcal{M}| + |\mathcal{T}| \times |\mathcal{M}|$ | $d \times 2r$ | $d \times 2r$ |
| MHR-$z$ | $d \times 2r \times |\mathcal{M}| + |\mathcal{T}| \times |\mathcal{M}| \times h$ | $|\mathcal{M}| \times h$ | $|\mathcal{M}| \times h$ |
| MHR | $d \times 2r \times |\mathcal{M}| + |\mathcal{T}| \times |\mathcal{M}| \times h$ | $d \times 2r \times |\mathcal{M}| + |\mathcal{M}| \times h$ | $d \times 2r$ |

Table 1: Number of parameters (per layer) used for each method. The calculation uses `LoRA` as the base adapter, modifying a linear transform in $\mathbb{R}^{d \times d}$. Note that the total number of parameters changed by `Full FT` is larger, given that the method also changes parameters for layers not modified by `LoRA`.

This is achieved by using a Gumbel-sigmoid distribution [Jang et al., 2017] during training, with $\hat{Z}_{\tau,j} \sim \texttt{Gumbel}(Z_{\tau,j})$. At each forward pass, `Poly` can be defined as :

$$\boldsymbol{A}^\tau = \sum_i \alpha_i \boldsymbol{A}^{(i)}; \ \boldsymbol{B}^\tau = \sum_i \alpha_i \boldsymbol{B}^{(i)} \tag{Poly}$$

where $\alpha_i = \frac{\hat{Z}_{\tau,i}}{\sum_j \hat{Z}_{\tau,j}}$ , and $\boldsymbol{A}^{(i)}, \boldsymbol{B}^{(i)}, \boldsymbol{A}^\tau, \boldsymbol{B}^\tau \in \mathbb{R}^{d \times r}$. We normalize the mixing coefficients $\hat{Z}_{\tau,i}$ for each task to ensure that the number of active modules does not affect the norm of $\boldsymbol{A}^\tau, \boldsymbol{B}^\tau$. Overall, this approach enables different subsets of *modules* to be activated for the current layer and combined in a task-specific way. Following LoRA, the output of the `Poly` layer is added to the output of the original layer of the frozen backbone: $\boldsymbol{h} = \boldsymbol{W}_0 x + s \boldsymbol{A}^\tau (\boldsymbol{B}^\tau)^\top x$.

During multi-task pre-training, for each query, key, value, and output projection in self-attention layers, the parameters learned by `Poly` are the adapter parameters, $\{\boldsymbol{A}_i, \boldsymbol{B}_i\}_{i=1}^{|\mathcal{M}|}$, and the routing matrices $\boldsymbol{Z}$. During fine-tuning, for each test task $\tau$, `Poly` randomly initialize the routing vector $z_\tau \in \mathbb{R}^{1 \times |\mathcal{M}|}$ and fine-tunes both $z_\tau$ and all the modules parameters $\mathcal{M}$.

## 3 Multi-Head Adapter Routing (`MHR`)

In `Poly`, module combination remains *coarse*: only linear combinations of modules are possible, and thus the resulting aggregated adapter remains a linear function of the modules. We propose to augment the expressivity of the module combination while keeping the parameter count similar. `MHR` (Fig. 1) takes inspiration from multi-head attention [Vaswani et al., 2017]: it partitions the input dimensions into $h$ different disjoint subsets, performs a separate `Poly`-style combination for each of them, and finally concatenates them. This corresponds to learning a different routing matrix $\boldsymbol{Z}$ for each subset of input features, therefore enabling the model to select different adapters for different subsets of the input dimensions. This aggregation approach is *piecewise* linear (i.e., linear within disjoint intervals), which allows for more expressive combinations of modules.

In each `MHR` layer, the routing function is a third-order tensor $\boldsymbol{\mathsf{Z}} \in \mathbb{R}^{|\mathcal{T}| \times |\mathcal{M}| \times h}$, where $\boldsymbol{\mathsf{Z}}_{:,:,h} \in \mathbb{R}^{|\mathcal{T}| \times |\mathcal{M}|}$ is a 2D slice of the tensor $\boldsymbol{\mathsf{Z}}$. A slice represents the routing matrix for each of the $h$ heads. Let us denote with $\boldsymbol{W}[k] \in \mathbb{R}^{\frac{d}{h} \times r}$ the $k$-th partition along the rows of the matrix $\boldsymbol{W} \in \mathbb{R}^{d \times r}$. The adapter parameters $\boldsymbol{A}^\tau \in \mathbb{R}^{d \times r}$ for task $\tau$, and for each adapter layer, are computed as (similarly for $\boldsymbol{B}^\tau$):

$$\boldsymbol{A}_k^\tau = \sum_j \boldsymbol{A}_j[k] \cdot \frac{\hat{Z}_{\tau,j,k}}{\sum_j \hat{Z}_{\tau,j,k}} \ \text{ with } \ \boldsymbol{A}_k^\tau \in \mathbb{R}^{\frac{d}{h} \times r}, \tag{MHR}$$
$$\boldsymbol{A}^\tau = \texttt{concat}(\boldsymbol{A}_1^\tau, \dots, \boldsymbol{A}_h^\tau)$$

where `concat` concatenates along the first dimension. Multi-task pre-training and fine-tuning are similar to `Poly`. Note that `MHR` results in only a negligible increase in the total amount of parameters, since most of the parameters are contained in the LoRA weights $\boldsymbol{A}, \boldsymbol{B}$ (Tab. 1).

**Routing-Only Fine-Tuning (`MHR-`$z$)**   Prior work [Shao et al., 2023, *inter alia*] has shown that compositional generalization can be achieved by learning to (re-)combine in novel ways pre-existing modules. We investigate whether fine-tuning the module parameters is really needed for few-shot adaptation in the context of both `Poly` and `MHR`. Therefore, we name `Poly-`$z$ and `MHR-`$z$ the variants that, during few-shot adaptation, keep the parameters of the modules learned during multi-task pre-training fixed and just update the routing parameters $\boldsymbol{Z}$. Crucially, this enables highly parameter-efficient adaptation: for `LoRA` adapters, $\boldsymbol{A}$ and $\boldsymbol{B}$ matrices constitute the overwhelming majority of parameters. Therefore, by freezing the $\boldsymbol{A}, \boldsymbol{B}$ matrices and only updating $\boldsymbol{Z}$, we can significantly reduce the parameter cost when transferring knowledge to a new task.

**Adapter Average Fine-Tuning (`MHR-`$\mu$)**   To assess the importance of the routing parameters during few-shot adaptation, we propose an additional variant of `MHR`, `MHR-`$\mu$, which shares the same multi-task pre-training procedure as `MHR`, but for each test task $\tau$, fixes $z_\tau = (1/|\mathcal{M}|, \ldots, 1/|\mathcal{M}|)$ during few-shot adaptation. This is equivalent to discarding the routing parameters and averaging the module parameters into a single one before fine-tuning. Specifically, the adapter used during fine-tuning is initialized with (similarly for $\boldsymbol{B}^\tau$):

$$\boldsymbol{A}^\tau = \frac{1}{|\mathcal{M}|} \sum_i \boldsymbol{A}_i^*; \ \boldsymbol{A}^\tau \in \mathbb{R}^{d \times r} \qquad \text{(MHR-}\mu\text{)}$$

where $\boldsymbol{A}_i^*$ are the parameters of the adapters after `MHR` multi-task pre-training. Note that, differently from `MHR`, `MHR-`$\mu$ fine-tunes the same amount of parameters as the single adapter baseline. Thus, any difference in performance between the single adapter baseline and `MHR-`$\mu$ comes from differences in the adapter initialization and must be due to the optimization process taking place in the multi-task pre-training, before few-shot adaptation.

**Routing Granularity**   In the original `Poly`, Ponti et al. [2023] showed that learning a routing matrix $\boldsymbol{Z}$ for each model layer gave better performance than sharing a single $\boldsymbol{Z}$ matrix across all layers. We therefore investigate whether this holds true also for its multi-head counterpart, `MHR`. In addition, we explore intermediate approaches between one $\boldsymbol{Z}$ per layer and a single one shared for the entire model. In particular, we consider sharing $\boldsymbol{Z}$ 1) for the adapter layers belonging to the same Transformer block; or 2) for every block of $l$ layers, which enables us to easily trade off expressivity for parameter efficiency. As we will demonstrate in section 5.1, this is an efficient mechanism to navigate this Pareto front in regimes of very small budgets of parameters per task.

# 4   Experiments

Our experimental evaluation aims to answer three research questions: 1) Does the expressivity of the routing function matter? 2) Why do routing-based PEFT methods yield superior performance? 3) Is routing useful during both multi-task pre-training and few-shot adaptation? We first present the baselines and datasets used in our evaluation and then discuss each question in turn.[1]

## 4.1   Baselines

In addition to `Poly`, we compare `MHR` to the following baselines for task-level generalization.

`LoRA/(IA)`[3] trains a single adapter common to all pre-training tasks, which is then fine-tuned on each test task separately. This is arguably the most widespread approach for parameter-efficient cross-task generalization [Liu et al., 2022, Pfeiffer et al., 2023].

`AdapterSoup` Chronopoulou et al. [2023] trains a different adapter for each task. The method only averages the adapter weights of the training tasks most similar to a given test task, before proceeding with few-shot adaptation. To compute task relatedness, we measure the cosine similarity of sentence embeddings for each task averaged over their training dataset. Notably, unlike the methods proposed in this paper, there is no knowledge sharing (nor interference) during multi-task pre-training as task adapters are trained independently.

---

[1]We note that all experiments were run on a single NVIDIA A100 GPU.

| T0 Dataset | Avg. Test |
|---|---|
| *Backbone T5-XL* | |
| (IA)$^3$ | $62.4_{0.4}$ |
| AdapterSoup | $62.1_{1.0}$ |
| LoRA | $66.0_{1.6}$ |
| LoRA-big | $65.4_{0.9}$ |
| Poly-$z$ | $66.4_{0.3}$ |
| Poly | $68.0_{1.0}$ |
| MHR-$z$ | $68.3_{0.8}$ |
| MHR | $\underline{69.1}_{1.0}$ |
| *Backbone T0-3B* | |
| T-Few Liu et al. [2022] | $66.2_{0.5}$ |
| AdapterSoup | $66.1_{0.6}$ |
| LoRA | $67.4_{0.8}$ |
| Poly-$z$ | $65.3_{1.0}$ |
| Poly | $69.0_{0.8}$ |
| MHR-$z$ | $68.4_{1.2}$ |
| MHR | $\underline{69.3}_{1.2}$ |

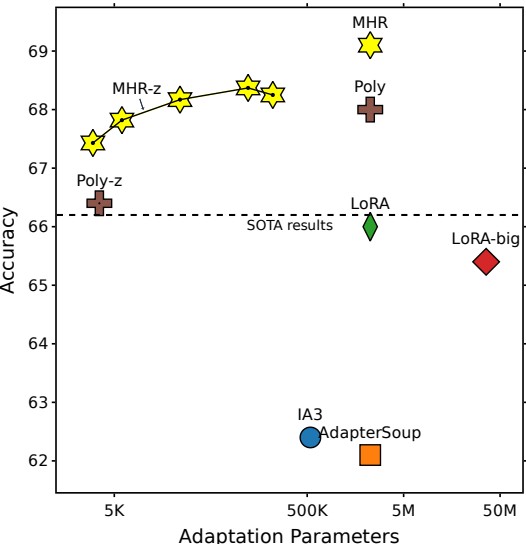

Figure 2: *Left:* Results of few-shot adaptation on T0 dataset Sanh et al. [2022]. We report the mean of the best validation accuracy for each test task. Subscripts correspond to standard deviation. *Right:* Accuracy of PEFT methods on the T0 dataset when applied on top of T5-XL. The x-axis shows the parameter count during the fine-tuning process.

## 4.2 Datasets

We test our methods on the T0 Sanh et al. [2022] evaluation suite, following the same setup as Liu et al. [2022], and SuperNI Wang et al. [2022a], a large-scale dataset with more than 1,600 training tasks.

**T0 Tasks**  We follow the pre-training and fine-tuning procedure discussed in Liu et al. [2022], using hyper-parameters and losses suggested in the public codebase for T-Few.[2]

All methods were tested with T5-XL Raffel et al. [2020] and T0-3B Sanh et al. [2022] as the backbone model. Crucially, T5 is simply pre-trained on (masked) language modelling, whereas T0 is further instruction tuned: in particular, the full model is fine-tuned on examples from multiple training tasks that have been augmented with task instructions. To ensure fairness for all methods, we report the median and standard deviation of the best validation accuracy for each test task across 3 seeds, when evaluated every 50 training epochs. We treat each data subset–template pair as a unique task, yielding a total of 313 tasks.

**SuperNI**  To limit computational costs, we report the result on 20 out of 119 test tasks. Tasks were chosen at random, with the requirement that at least 300 examples were available, and were equally split into 100 training, 100 validation and 100 test examples. For every method, we perform early stopping on the validation set. We report results with Rouge-L averaged across 3 seeds. All methods use T5-XL [Raffel et al., 2020] as the backbone and not T0, as T0 training tasks and SuperNI test tasks may overlap.

## 5 Results and Discussion

### 5.1 Does the expressivity of the routing function matter?

**MHR outperforms PEFT approaches**  We start our analysis by evaluating the effectiveness of our proposed technique when applied over a backbone that has not undergone prior training on

---

[2]https://github.com/r-three/t-few

instruction-following data (T5-XL). As indicated in the T0 benchmark results in the top table of Fig. 2, it is clear that multi-head routing techniques have a distinct advantage, outperforming both single-head routing `Poly` by **1.1%**, and surpassing standard `LoRA` approaches by an impressive **3.1%**. We also study the impact of performing instruction tuning of the full backbone before adapter training. To this end, we also experiment with T0-3B as a backbone. In the bottom table of Fig. 2, we can observe that while the relative gap between `MHR` and baselines is smaller, multi-head routing still manages to yield favourable results. Hence, the gains of `MHR` compound with other multi-task methods such as instruction tuning. Finally, we turn our attention towards the SuperNI dataset (Tab. 2). Here, `MHR` continues to surpass analogous baselines.

**MHR-$z$ facilitates extreme parameter efficiency** Fig. 2 (right) reveals intriguing findings regarding `MHR-$z$`. When we restrict training to only the routing parameters $Z$ in the original `Poly`, the results are unfortunately not up to par with its version where both routing and adapters are updated. However, when we apply the same constraint to `MHR`, the performance is significantly closer to the optimum achieved in this setting. In fact, `MHR-$z$` surpasses prior baselines while simultaneously necessitating fewer parameters for effective adaptation to new tasks. Moreover, by controlling the number of layers which share the same $Z$ allocation (see sec. 3), `MHR-$z$` is able to trade-off performance for parameter efficiency, even surpassing `Poly-$z$` in settings with only 3K trainable parameters per test task (see also § A.2.1 for a more in-depth analysis). This trend is similarly observed in the SuperNI benchmark (Tab. 2), where updates restricted to the routing parameters yield performance on par with standard fine-tuning. We therefore conclude that the `MHR-$z$` represents a robust approach for achieving extreme parameter efficiency in adaptation.

**Additional routing heads is more beneficial than extra modules** In the original `Poly` approach, a tradeoff between capacity and parameter efficiency can be achieved by adding extra modules for each adapter layer. However, this results in a linear increase in the number of multi-task parameters, which can become impractical. To explore a more effective tradeoff, we investigate the option of adding additional routing heads instead of extra modules. Fig 3 (right) presents the comparison between the two approaches. It demonstrates that increasing the number of routing heads leads to better performance compared to adding more modules. Importantly, the benefit of multi-head routing is twofold:

| SuperNI Dataset | Rouge-L |
|---|---|
| LoRA | $67.6_{0.8}$ |
| LoRA-big | $67.2_{0.7}$ |
| Poly-$z$ | $64.6_{0.3}$ |
| Poly | $67.8_{0.8}$ |
| MHR-$z$ | $68.0_{0.2}$ |
| MHR | $\underline{68.5}_{0.3}$ |

Table 2: Results on **SuperNI** dataset. Subscripts are standard deviation.

it provides increased expressivity for the model, while also maintaining parameter efficiency. This finding highlights the advantage of multi-head routing as a more effective approach for balancing expressivity and parameter count in few-shot adaptation scenarios.

**Routing-based methods also excel at the 11B scale** We proceed to evaluate if `Poly` and `MHR` surpass established PEFT approaches when trained over a larger model backbone. To accomplish this, we employ the 11B version of T0. As depicted in Tab. 3, routing-based methods once again outshine standard adapter training, surpassing our reproduction of the previous state-of-the-art in Liu et al. [2022] by over 2%. We observe that `Poly` and `MHR` show similar performance in standard fine-tuning, but `MHR` $z$-tuning remains more performant in routing-only fine-tuning. Indeed, `MHR-$z$` (221K params) outperforms `Poly-$z$` (3.5K params) by 2.9%, while still remaining more parameter efficient than Liu et al. [2022] (1.1M params).

| T0 Dataset | Avg. Test |
|---|---|
| *Backbone T0-11B* | |
| T-Few Liu et al. [2022] | $72.5_{0.9}$ |
| LoRA | $72.3_{1.0}$ |
| Poly-$z$ | $70.0_{0.6}$ |
| Poly | $\underline{74.9}_{0.6}$ |
| MHR-$z$ | $72.9_{0.8}$ |
| MHR | $74.7_{0.6}$ |

Table 3: Few-shot results over 11B parameter backbones.

## 5.2 Why do routing-based PEFT methods yield superior performance?

While our proposed methods have demonstrated promising results across a broad spectrum of datasets and varying adaptation parameter budgets, the question of *why* routing-based PEFT exhibits superior

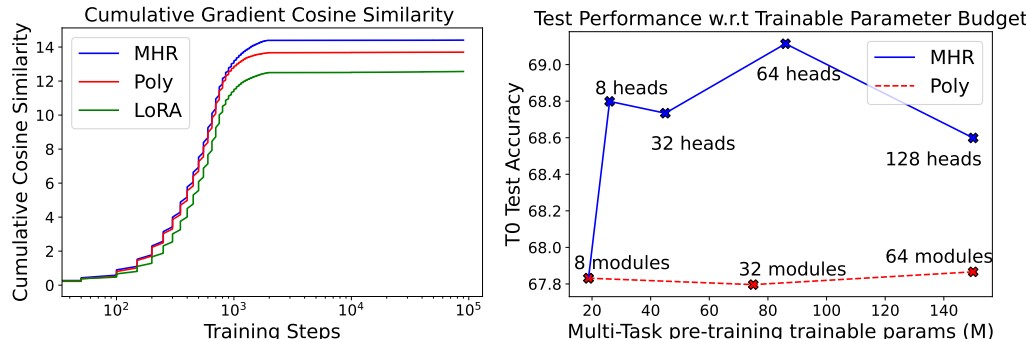

Figure 3: *Left:* Gradient alignment between tasks during multi-task pretraining. *Right:* Increasing the number of heads offer better scaling properties than increasing the number of modules.

performance remains unanswered. In this section, we aim to uncover the key components that drive `MHR`'s superior performance.

**Learning the Routing Function is essential** Given that `Poly` and `MHR` have access to more parameters than standard adapters during multi-task pretraining, we investigate whether this, and not the routing mechanism, is responsible for their superior performance. To do so, we compare them to a baseline approach. Instead of learning the routing function, we randomly assign a binary module allocation to each data point in a minibatch, disregarding task information. This random routing approach, akin to Wang et al. [2022b], allows us to directly assess the influence of additional parameters during multi-task training. At test time, the learned modules are averaged into a single one before fine-tuning; we therefore refer to this baseline as `Random-`$\mu$. On the T0 benchmark with the T5-XL backbone, `Random-`$\mu$ performs similarly to a standard LoRA adapter (66.0%), while `Poly` and `MHR` outperform it by **2%** and **3.1%** respectively. Therefore, we conclude that learning a routing function is crucial, and merely increasing capacity during training does not directly lead to improvements.

**`MHR` fosters transfer and mitigates interference across pretraining tasks** Recognizing the pivotal role of the multi-task pretraining step in bolstering `Poly`'s performance, we explore the extent of transfer and interference across training tasks. By monitoring the average gradient alignment for each task pair (in terms of cosine similarity) throughout the training process, we are able to gauge the level of positive transfer. As Fig. 3 (left) shows, `MHR` displays a greater degree of gradient cosine similarity across tasks compared to other PEFT alternatives, including `Poly`. This finding suggests that the enhanced flexibility offered by multi-head routing may serve to mitigate interference across tasks to a larger extent than standard routing while simultaneously promoting positive transfer.

### 5.3 Is routing important for task generalization?

We assessed the importance of routing during pretraining. We now proceed to verify whether it is important to learn routing during few-shot adaptation, too. `Poly-`$\mu$ and `MHR-`$\mu$ consistently outperform `LoRA`, and match the performance of `Poly` / `MHR` (Tab. 4). This demonstrates that, for few-shot adaptation, the average of the pre-trained modules provides a better initialization than learning an adapter shared across all the tasks during pre-training. The consistently superior performance of `Poly-`$\mu$ with respect to `Random-`$\mu$ and `AdapterSoup` stresses the importance of routing during multi-task pre-training (but not during adaptation), which provides an effective adapter initialization for few-shot learning. This

| T0 Dataset | Test Acc. |
|---|---|
| LoRA | $66.0_{1.6}$ |
| AdapterSoup | $62.1_{1.0}$ |
| Poly | $68.0_{0.8}$ |
| Poly-$\mu$ | $67.8_{0.6}$ |
| MHR | $69.1_{1.1}$ |
| MHR-$\mu$ | $69.1_{0.9}$ |

| SuperNI | Rouge-L |
|---|---|
| LoRA | $67.6_{0.8}$ |
| Poly | $67.8_{0.8}$ |
| Poly-$\mu$ | $68.3_{0.5}$ |
| MHR | $68.5_{0.6}$ |
| MHR-$\mu$ | $68.5_{0.8}$ |

Table 4: Evaluating the impact of modular adaptation at test time.

| Method | Zero-Shot Test on 11 Held-out Tasks | | | |
|---|---|---|---|---|
| | $k = 0$ | $k = 1000$ | $k = 5000$ | $k = 10000$ |
| *Backbone T5-XL* | | | | |
| LoRA | 56.5 | 56.0 | 56.1 | 55.7 |
| Poly-$\mu$ | 46.0 | 53.0 | 56.8 | 56.3 |
| MHR-$\mu$ | 48.0 | 58.0 | 57.1 | 56.3 |
| *Backbone T0-11B* [Sanh et al., 2022] | 61.0 | | | |
| LoRA | 61.2 | 61.6 | 61.5 | 61.5 |
| Poly-$\mu$ | 62.1 | 63.6 | 63.9 | 64.4 |
| MHR-$\mu$ | 63.5 | 64.5 | 64.5 | 64.4 |

Table 5: Zero-shot performance for MHR and the baselines, reported as the average over the 11 evaluation datasets from Sanh et al. [2022]. To obtain these zero-shot results, we average the learnt Poly/MHR adapters before performing $k$ additional fine-tuning steps on the multi-task pretraining data. This effectively enables zero-shot transfer to downstream tasks using the same amount of parameters/flops as the baseline LoRA. MHR outperform baseline LoRA by up to 3% absolute accuracy points on T0-11B.

finding could potentially inspire future work for improving meta-learning and weight-averaging approaches [Izmailov et al., 2018].

**MHR-$\mu$ excels at zero-shot learning**   For many downstream tasks of interest, additional labelled data may not be available. In such settings, it is unclear how to leverage MHR-$\mu$ and Poly-$\mu$ methods. To address this, we fine-tune the average of the multi-task trained adapters on the multi-task pre-training data (instead of using the downstream few-shot data), for an additional $k$ steps. The results are presented in Table 5. We find that without any additional fine-tuning ($k = 0$), averaging the adapters does not yield good results. This is due to a potential mismatch between adapters learned via task-specific routing, and the uniform routing strategy. We can observe that when fine-tuning the average of the adapters on the multi-task pre-training data for an additional $k$ steps, MHR-$\mu$ show strong performance when evaluated in a zero-shot manner. For a fair comparison, we also additionally fine-tune LoRA for the same number of additional steps. Our best model achieves a zero-shot performance of 64.5 on top of T0-11B, achieving an absolute gain of 3.5% accuracy points.

## 6   Related Work

Multi-task learning is effective for low-resource tasks [Wei et al., 2022, Aribandi et al., 2022, Sanh et al., 2022], as knowledge can be borrowed from similar tasks by sharing the model parameters. Multi-task learning has also been applied across languages and modalities [Ponti et al., 2019, Bugliarello et al., 2022]. In the context of NLP, several families of methods enable learning new tasks from a limited set of labelled examples. Few-shot in-context learning [ICL; Brown et al., 2020], where examples of a new task are concatenated into an input prompt, enables models to generalize to *unseen* tasks without any gradient-based training. Such approaches are however sensitive to the prompt format and example ordering [Zhao et al., 2021]. More importantly, ICL methods incur a significant compute overhead, as for every prediction, the full set of examples must be processed by the model [Liu et al., 2022]. To remedy this, many parameter-efficient fine-tuning (PEFT) methods have been proposed as an alternative to ICL, where a small number of new parameters are added over the frozen pretrained network. To name a few, LoRA [Hu et al., 2022] injects learnable low-rank matrices into each Transformer layer. Alternatively, the learnable matrix can be sparse, selecting nonzero shifts via the Lottery-Ticket hypothesis [Ansell et al., 2021] or via their approximate Fisher information [Sung et al., 2021]. Finally, prefix-tuning methods [Li and Liang, 2021] prepend learnable embeddings to the input or intermediate representations to specialize the model towards a downstream task.

Modular networks partition their parameters into several expert modules, each of them specialized to handle specific sub-tasks [Jacobs et al., 1991, Kirsch et al., 2018]. Modular networks are an appealing solution to the problem of adapting to unseen tasks [Corona et al., 2021], as the model

can leverage its existing modules and recombine them in a novel way, thus achieving systematic generalization [Bahdanau et al., 2019]. They have also been tested in learning scenarios with data presented sequentially [Ostapenko et al., 2021], and with changing environments Goyal et al. [2021]. In NLP, mixture-of-experts (MoE) models [Shazeer et al., 2017, Fedus et al., 2022], where a learned gating mechanism routes token representations to appropriate experts (Feed-Forward layers), have shown success in scaling the number of parameters while retaining time efficiency. This results in higher performance when compared to their dense counterparts using a similar compute budget.

## 7   Conclusions

In this paper, we tackle the challenge of generalizing to new tasks based on a few examples after multi-task pre-training. Specifically, we focus on Polytropon [Ponti et al., 2023], a model where each task is associated with a subset of adapters by a routing function. We investigate how varying the level of control afforded by the routing function impacts performance on two comprehensive benchmarks for multi-task learning, T0 and Super-Natural Instructions. First, a newly proposed variant of the routing function, where multiple heads are responsible for different subsets of input dimensions, improves consistently over all other baselines, including LoRA and $(IA)^3$ adapters. Second, we identify the cause of the success of routing in its ability to prevent interference between tasks, as it yields a better alignment between their gradients. Third, we find that simple averaging of all multi-task pre-trained adapters before few-shot adaptation to new tasks provides comparable performance, thus offering state-of-the-art performance for single-adapter few-shot learning. Multi-head routing demonstrates the importance of fine-grained adapter selection for sample-efficient generalization and holds promise to improve other modular methods, such as Mixtures of Experts [MoEs; Fedus et al., 2022] in future research.

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

# A Appendix

## A.1 Additional Results

More detailed numbers on the T0 Sanh et al. [2022] and SuperNI Wang et al. [2022a] datasets using different backbones, and different adapter layouts over the base model are found in Table 6. `Multi-Task params` is the number of additional parameters that must be conserved after

| Model | Multi-Task Params | Adaptation Params | Avg. Test |
|---|---|---|---|
| **T0 Dataset** | | | |
| *Backbone T5-XL-LM* | | | |
| `Multi-Task Full Finetuning + LoRA` | 2.8B | 2.2M | $68.9_{x.x}$ |
| $(IA)^3$ | 540K | 540K | $62.4_{0.4}$ |
| `AdapterSoup` | 84M | 2.2M | $62.1_{1.0}$ |
| `LoRA` | 2.2M | 2.2M | $66.0_{1.6}$ |
| `LoRA-big` | 35M | 35M | $65.4_{0.9}$ |
| `Poly`-$z$ | 17M | 3.5K | $66.4_{0.3}$ |
| `Poly` | 17M | 2.2M | $68.0_{1.0}$ |
| MHR-$z$ *(64 h)* | 17M | 220K | $68.3_{0.8}$ |
| MHR *(64 h)* | 17M | 2.2M | $\underline{69.1}_{1.0}$ |
| *Backbone T0-3B* | | | |
| `T-Few` Liu et al. [2022] | 540K | 540K | $66.2_{0.5}$ |
| `AdapterSoup` | 84M | 2.2M | $66.1_{0.6}$ |
| `LoRA` | 2.2M | 2.2M | $67.4_{0.8}$ |
| `LoRA-big` | 35M | 35M | $68.0_{0.8}$ |
| `Poly`-$z$ | 17M | 3.5K | $65.3_{1.0}$ |
| `Poly` | 17M | 2.2M | $69.0_{0.8}$ |
| MHR$z$ *(64 h)* | 17M | 220K | $68.4_{1.2}$ |
| MHR *(8 h)* | 17M | 2.2M | $\underline{69.3}_{1.2}$ |
| *Backbone T0-3B light* version : (`k, v, ff` layers only) | | | |
| *l-*LoRA *(rank 1)* | 934K | 934K | $66.2_{0.9}$ |
| *l-*LoRA *(rank 16)* | 15M | 15M | $67.6_{1.1}$ |
| `AdapterSoup` (*l-*LoRA) | 35M | 934K | $64.9_{1.0}$ |
| *l-*`Poly`-$z$ | 7.5M | 2.1K | $62.9_{1.2}$ |
| *l-*`Poly` | 7.5M | 934K | $68.0_{0.5}$ |
| *l-*MHR$z$ *(32 h)* | 7.5M | 74K | $66.8_{1.1}$ |
| *l-*MHR *(8 h)* | 7.5M | 934K | $\underline{68.5}_{0.7}$ |
| **SuperNI Dataset** | | | Rouge-L |
| *Backbone T5-XL-LM light* version : (`k, v, ff` layers only) | | | |
| *l-*LoRA | 934K | 934K | $67.6_{0.8}$ |
| *l-*LoRA-big | 18M | 18M | $67.2_{0.7}$ |
| *l-*`Poly`-$z$ | 7.5M | 2.1K | $64.6_{0.3}$ |
| *l-*`Poly` | 7.5M | 934K | $67.8_{0.8}$ |
| *l-*MHR$z$ *(64 h)* | 7.5M | 147K | $68.0_{0.2}$ |
| *l-*MHR *(8 h)* | 7.5M | 934K | $\underline{68.5}_{0.3}$ |

Table 6: (top) Results on T0 dataset Sanh et al. [2022], we report the mean of the best validation accuracy for each test task, when evaluated every 50 train epochs. `T-Few` is our reproduction of the results in Liu et al. [2022]. `LoRA-big` means a LoRA adapter with a larger rank. (bottom) Results on SuperNatural Instructions dataset.

multi-task pretraining to enable transfer to a downstream task. `Adaptation Params` refer to the number of parameters required to learn a new downstream task. For e.g. `Poly` and `MHR`, the multi-task parameters includes the learned modules, but not the routing over the training tasks, as these are not required for transfer on a new task. Moreover, variants which average the learned modules prior to fine-tuning (`MHR`-$\mu$ and `Poly`-$\mu$) will have both multi-task and adaptation parameters equal to that of a single shared adapter, since after multi-task pretraining one can average the modules.

## A.2 Navigating the parameter efficiency / performance trade-off of tuning only the routing

Here we provide additional results on how different routing based methods can be more expressive when only learning a new routing function (over *frozen* modules) to adapt to a new task.

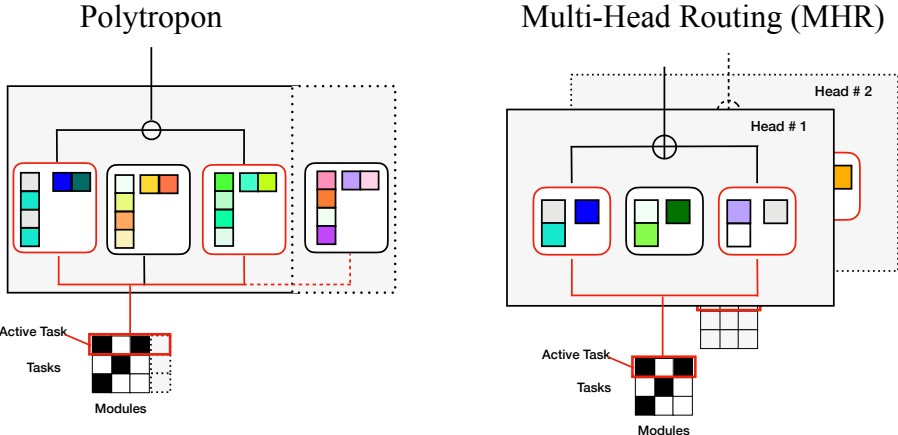

Figure 4: Different ways to control the expressivity of routing based methods. *Left :* In Polytropon, one can only add additional modules, resulting in a linear parameter increase. *Right :* In MHR, additional heads only introduce routing matrices $Z$, resulting in a negligible parameter increase.

In Fig. 4 (left), we see that in order to build more expressive routing functions $Z$, in Poly one can only do so by increasing the number of skills at each layer. However, this has a significant impact on the number of multi-task parameters which much be kept in order to perform few-shot transfer. MHR on the other hand, can increase routing capacity in a much more parameter efficient way.

### A.2.1 On the granularity of routing tensor in MHR

Here we provide additional results when modifying the granularity of $Z$ for MHR. We see that one can easily trade-off more parameters for better performance.

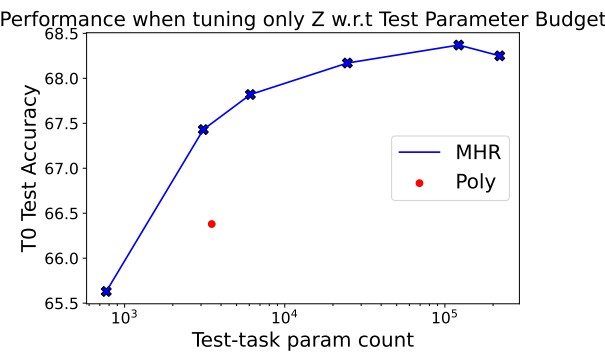

Figure 5: Routing-Only Fine-Tuning (MHR-$z$)

## B Broader Impact

In our work, we focus on advancing parameter-efficient fine-tuning methods for cross-task generalization. While our research primarily addresses technical challenges and performance improvements, when applying such methods, it is crucial to consider the potential negative societal impacts. Specifically, we believe that prior to applying our proposed adaptation method, critically examining the potential biases and ethical implications of the underlying large language model, and the data itself must be properly addressed. This includes issues related to fairness, privacy, and the spread of misinformation.

