# OpenReview forum: "Multi-Head Adapter Routing for Cross-Task Generalization"
_NeurIPS.cc/2023/Conference — NeurIPS 2023 poster_

### Official Review · Reviewer_m1AN · 2023-07-02

**Soundness:** 3 good
**Presentation:** 3 good
**Contribution:** 3 good
**Rating:** 5
**Confidence:** 3

**Summary:**

This paper proposed a new parameter-efficient few-shot fine-tuning method for pretrained language models. The method is a follow-up work of Poly. The authors proposed to fine-tuning the routing function and freezing the multi-head adapters. This way, the number of updated parameters reduced significantly while achieving a similar accuracy level in the down-streaming tasks.

**Strengths:**

- This paper studied an important problem in efficient fine-tuning of a pretrained language model, i.e., how to achieve a better trade-off between updated parameters and the final accuracy. The paper is well-motivated.
- Overall the paper is well-written, especially the experimental section. The authors conducted a comprehensive ablation study on some key questions raised in the paper.
- Although the method adopted in the paper (updating routing parameters only) was simple and straight forward, it seemed to work well in multiple different down-streaming tasks.

**Weaknesses:**

- The main shortage of the paper is the limited novelty. I think in comparison to Poly, the main difference of this work is to fine-tune routing only without the adapters. The authors conducted some heuristic analysis in the experimental section to understand the intuition behind. However, the paper may still not distinguish quite clear to the prior work. I would like to see more quantitative results on why fine-tuning routing is essential and necessary, e.g., the learned multi-head adapters are orthogonal in the task space?

- Some method parts are not fully clear to me. For example, the authors discussed (IA)^3 in Sec.2.1. However, the definition of h^{k,v} and h^f are not properly introduced.



**Questions:**

See the weakness

**Limitations:**

Yes

---

> ### Author Rebuttal · Authors · 2023-08-08
>
> Thank you for your feedback. We address below your concerns point by point.
>
> #### **1. On the novelty of the proposed method**
> We refer the reviewer to the global comments.
>
> #### **2. On why finetuning (only) the router is necessary**
> We do not mean to claim that fine-tuning only the router is necessary. However, we show that, contrary to Poly, MHR offers this possibility: one of the motivations for fine-tuning only the router is to enable higher parameter efficiency for new tasks. If we misunderstood your question, please let us know and we will be happy to engage.
>
> #### **3. Further clarifications**
> We apologize for not properly introducing $h^{k,v}$ and $h^f$. These vectors represented the output of the the keys and values in attention mechanisms and the inner activations in position-wise feed-forward networks.

---

> > ### Comment · Reviewer_m1AN · 2023-08-14
> >
> > Thanks for the authors' response.
> > I think overall this paper had a solid experimental section and might benefit the community about model adaptation although it had a limited novelty. I keep my original rating (Borderline accept).

---

### Official Review · Reviewer_1NFM · 2023-07-02

**Soundness:** 4 excellent
**Presentation:** 3 good
**Contribution:** 3 good
**Rating:** 7
**Confidence:** 4

**Summary:**

The authors propose a new parameter-efficient routing method, Multi-head routing (MHR), which combines parameter subsets as opposed to averaging all weights together. They find this yields better performance, and even only finetuning the routing matrix after initial training works well. Furthermore, they explore the training dynamics of MHR and find its gradients are more aligned than other approaches, suggesting that routing-based training helps to mitigate negative transfer during multi-task training.

**Strengths:**

A straightforward approach, with well-performed experiments over a wide variety of tasks. Relaxing the routing to allow for different parameter subsets is a nice extension of the Poly work. Results seem consistent across settings. The exploration of why these routing approaches work well is interesting and poses some interesting questions that could be explored by future work.


**Weaknesses:**

- I think the adaptersoup baseline is a bit unfair since it uses a different parameter-efficient adaptation method to MHR/Poly. Ideally, you would apply the AdapterSoup approach to these other methods. Otherwise, it’s hard to say if adaptersoup is truly worse or if it is just that adapters are suboptimal compared to LoRA.
- Gains are somewhat small over Poly, but this can be the case for parameter-efficient tuning. It would be useful to compute statistical significance.
- The finding that gradient cosine similarity is enhanced is interesting, but is weak evidence (in my opinion) for task interference/transfer. The suggestion that MHR is helping with negative/positive transfer would be much better served with some experiment directly targeting this (e.g. examining MHR with two tasks known to cause positive/negative transfer, and examining the gains/gradients there)
- The fact that using routing doesn't help in the few-shot stage (section 5.3) suggests that averaging and throwing away the modules learnt via MHR would be a better strategy than keeping the routing, no? Since keeping one set of PEFT weights and no routing is simpler than continuing the routing strategy.

Overall, I think this paper is solid and well done. While the gains of the method are small, I think the idea is interesting and the results around routing-only finetuning and task gradient alignment are interesting and pose interesting questions for future works.


**Questions:**

- Do you think the adaptersoup baseline is hindered by its use of adapters? What would its performance be like if you used Lora as the PEFT module instead?
- If averaging the learnt modules is better than routing during few-shot adaptation, why should we use the routing method after the multitask pretraining stage? Isn’t this simpler than the main MHR approach proposed?
- Which T5 XL variant did you use? It would be useful to specify this, since some T5 variants had multi-task data inserted during pretraining (I’m assuming you used the v1.1 lm-adapted ones, which did not).


**Limitations:**

The authors do not explicitly discuss limitations, although they discuss impacts in the appendix. It would be good to directly add a section on limitations, including e.g., the need to perform training over few-shot data vs using an ICL approach, the fact that routing is not needed during few-shot training (it seems), or any other limitations you can think of.

---

> ### Author Rebuttal · Authors · 2023-08-08
>
> We thank the reviewer for providing a thorough and constructive review. We will address the questions one by one.
>
> #### **1. AdapterSoup Baseline and Backbone Clarifications**
> We apologize for the lack of clarity regarding this. Indeed, our AdapterSoup baseline uses the same LoRA configuration (same base layers, same LoRA rank). As you pointed out, in order to properly assess the contribution of the routing mechanism of AdapterSoup, we kept other design choices fixed. Moreover, you are correct in our choice of backbone, we opted for the t5-xl-lm-adapt v1.1. Lastly, we agree with the limitations you raised regarding ICL vs adapter tuning. We will add this to the next version.
>
> #### **2. On whether to route vs average during-few shot adaptation**
> We agree that in settings where transfer is limited to a limited set of test tasks, averaging the learned modules is a more straightforward solution. However, in settings where high levels of parameter efficiency are required per task, options such as MHR-z require that the skills be kept. Moreover, we believe that an interesting future research direction would be to apply MHR to continual learning settings, where modular methods have been shown to work well [3]. In such setting, the multi-task optimization step could be composed of multiple phases, in which case keeping the full set of skills may be beneficial.
>
> #### **3. On statistical efficiency and pertinence of results**
> We refer the reviewer to the global comments.
>
> #### **4. On gradient alignment as a proxy for transfer / interference during training**
> We agree with the reviewer that indeed, a more thorough investigation on how MHR effects aids transfers / mitigates interference would be beneficial. While we could not find clear task pairs which are known to cause interference, we can however look at alignment across tasks known to be similar. For this, we looked at the cosine similarity of the resulting adapters for all 36 **summarization tasks**. We found that MHR had an average alignment of **0.76** for MHR and **0.73** for Poly.  We repeated this experiment for all **question answering tasks**, and again found superior alignment for MHR than Poly (**0.77** vs **0.74**).
>
> [3] Ostapenko, Oleksiy, et al. "Continual learning via local module composition." Advances in Neural Information Processing Systems 34 (2021): 30298-30312.

---

> > ### Comment · Reviewer_1NFM · 2023-08-14
> > **Re: Rebuttal**
> >
> > Hi, thank you for the response and clarifications! I've carefully read your response and the other reviews and am satisfied and keeping my score as-is. I agree that the multi-head splitting over the parameters of the PEFT method itself is interesting and novel, and I think that the experiments and ablations provide interesting insight into multi-task and parameter-efficient learning. While it is somewhat incremental (the overall framework not being *that* different to Poly), I still think the paper provides useful and interesting findings for the field, and passes my bar for novelty.
> >
> > Following up on the response:
> >
> > > For this, we looked at the cosine similarity of the resulting adapters for all 36 summarization tasks. We found that MHR had an average alignment of 0.76 for MHR and 0.73 for Poly. We repeated this experiment for all question answering tasks, and again found superior alignment for MHR than Poly (0.77 vs 0.74).
> >
> > What was the average/median alignment across all tasks, and between diff-task pairs (e.g. summarisation/question-answering)? On their own, it's hard to gauge if the difference between MHR and Poly here is significant, and if the alignment values here are actually high compared to orthogonal or diverging tasks.

---

> > > ### Author Response · Authors · 2023-08-15
> > >
> > > Thank you for taking the time to engage with us. Here are some clarifications regarding the adapter alignment results shared in the previous reply. We report the average adapter cosine similarity across different combinations of tasks.
> > >
> > > | Task Pairs                          | MHR  | Poly |
> > > |-------------------------------------|------|------|
> > > | all `(313 * 312 / 2)` task pairs      | 76.3 | 74.1 |
> > > | all Q/A task pairs                  | 76.7 | 74.3 |
> > > | all summarization task pairs        | 76.4 | 73.4 |
> > > | all Q/A vs summarization task pairs | 75.7 | 72.1 |
> > >
> > > While the gap across different task pairs is relatively small, we do see that overall, MHR tends to offer better alignment across similar (and less similar) task pairs. That being said, we agree with the reviewer that a more thorough investigation is needed to properly assess how MHR aids transfer across tasks. We will update the paper accordingly in the next version.
> > >
> > > Thank you.

---

### Official Review · Reviewer_5omF · 2023-07-04

**Soundness:** 3 good
**Presentation:** 2 fair
**Contribution:** 2 fair
**Rating:** 5
**Confidence:** 3

**Summary:**

This paper introduces a method called Parameter-efficient Fine-tuning (PEFT) to improve how pre-trained language models adapt to new tasks. They use small adapters and a routing function to select specific adapters for each task. The authors found that finer-grained routing provides better results and propose a method called Multi-Head Routing (MHR) that outperforms previous approaches. They also discovered that the success of their method is mainly due to improved multi-task optimization rather than specific adapter properties. They introduce a simplified variant called MHR-μ that achieves competitive performance with fewer parameters by discarding routing during fine-tuning.

**Strengths:**

++ The paper demonstrates clear and concise writing, making it easily comprehensible.

++ The authors conducted a thorough ablation study to thoroughly evaluate the effectiveness of their proposed approach.

**Weaknesses:**

-- The paper lacks significant technical novelty. The approach of decomposing the parameters in LoRA into different sets is not particularly interesting. Additionally, the designed modules bear strong resemblance to multi-head self-attention and MoE (Mixture of Experts). Moreover, the intuition behind these designs seems ad-hoc and lacks novel insight, which is crucial for assessing the value of the paper.

-- The paper fails to provide specific details about the model parameters used in the experiments. Considering that the proposed model is substantially larger than the baselines, it is necessary to have more information in order to properly evaluate the proposal.

**Questions:**

The authors are encouraged to address the mentioned concerns in order to provide a better understanding of their work. It would be beneficial for them to provide further clarification regarding the technical novelty of their proposed parameter decomposition approach in LoRA, highlighting any unique aspects that distinguish it from existing methods such as multi-head self-attention and MoE. Offering additional insights and motivations behind their design choices would also help in assessing the novelty and value of their work.

Furthermore, it is important for the authors to address the lack of specific information about the model parameters used in their experiments. Providing details about the size and configuration of the proposed model in comparison to the baselines would enable a more accurate evaluation of its effectiveness. These additional facts would assist in making a fair assessment and potentially lead to a reconsideration of the paper's overall score.

**Limitations:**

-- No limitation is discussed in the paper.

---

> ### Author Rebuttal · Authors · 2023-08-08
>
> We thank the reviewer for their feedback. We address the raised concerns point by point.
>
> #### **1. On the novelty of the proposed method**
> We refer the reviewer to the global comments
>
> #### **2. Additional information about model parameters and configuration**
> For all experiments in the paper, all the reported methods besides ia3 use LoRA adapters, and are inserted **on exactly the same layers**. In other words, the amount of adapted layers are the same. In the main results of the paper, we add adapters to key, value, query and output linear mappings of attention mechanisms, as well as to the two linear layers in the feed-forward transformer block. Except for “LoRA-big”, all methods using LoRA use a rank of 1.
>
> We refer to the additional parameters that must be conserved after multi-task pretraining as `Multi-Task Params`. We denote the additional parameters that must be kept for each new downstream task as the `Adaption Params`. These results are using T5-XL, the lm-adapted version with 3B params. Importantly, **MHR outperforms other baselines with more parameters** (see LoRA rank 16 and AdapterSoup).
>
> | Model             | Multi-Task Params | Adaptation Params | Avg. Test Performance |
> |--------------------|-------------------|-------------------|-----------------------|
> | IA3                 | 540K              | 540K              | 62.4                 |
> | AdapterSoup  | 84M               | 2.2M              | 62.1                 |
> | LoRA              | 2.2M              | 2.2M              | 66.0                 |
> | LoRA rank 16 | 35M               | 35M               | 65.4                 |
> | Poly-Z            | 17M                | 3.5K              | 66.4                  |
> | Poly               | 17M                | 2.2M              | 68.0                 |
> | Poly-mu         | 2.2M               | 2.2M              | 67.8                 |
> | MHR-z           | 17M                | 220K              | 68.3                 |
> | MHR              | 17M                | 2.2M              | 69.1                 |
> | MHR-mu        | 2.2M                | 2.2M             | 69.1                 |
>
> We see that methods with MHR routing better optimize the adaptation parameter / test set performance tradeoff. Additional results with different configurations can also be found in the Appendix A.1.
>
>
> Please let us know if you have any other questions.

---

> > ### Comment · Reviewer_5omF · 2023-08-15
> > **Borderline accept**
> >
> > I'd like to express my gratitude to the authors for addressing my concerns, particularly regarding the parameter size. I believe the paper presents a compelling approach to efficient fine-tuning and demonstrates notable performance improvements. As a result, I've adjusted my rating to borderline accept.

---

### Official Review · Reviewer_HX6c · 2023-07-07

**Soundness:** 3 good
**Presentation:** 4 excellent
**Contribution:** 3 good
**Rating:** 6
**Confidence:** 4

**Summary:**

In this work, the authors proposed MHR for cross-task generalization. To achieve extreme parameter efficiency, MHR- z and MHR-u are proposed to balance the performance and efficiency. Besides, this work emphasized the importance of the routing function, which is very insightful for the community.

**Strengths:**

1. Excellent presentations for motivation, and experimental results.

2. I am impressed by the performance of MHR-z, the accuracy is close to poly while there are only very few parameters that need to be adjusted in the fine-tuning stage.

**Weaknesses:**

1. The improvement of the MHR is quite limited, which only outperforms the Poly by 1.1%. Besides, there is no significant test, which is very important for such a marginal improvement.

2. Fig. 1 is not well presented.


**Questions:**

1. In Figure 3, why MHR with 32 heads performs worse than 8 heads version?

2. In Figure 1, I would suggest the authors to mark where is the task-specific head. In addition, More information is needed for MHR-Z in the Fig. 1 as well.

3. Does each task associated with a task-specific Z matrix? If not, how to select active modules based on the input task?

**Limitations:**

The authors did not present any limitations in the manuscript.
For the performance issue I pointed out in the weakness section, one potential reason could be the subheads are combined via averaging in your case. In some Mixture-of-experts papers, a set of dynamic weights can be generated to combine the experts dynamically based on the input information(task in your case).  This could be helpful to further improve the performance, and related discussion can be conducted in the manuscript as well.

[1] Condconv: Conditionally parameterized convolutions for efficient inference, NeurIPS 2019
[2] A mixture of h−1 heads is better than h heads, ACL 2020
[3] Attention over Self-attention: Intention-aware Re-ranking with Dynamic Transformer Encoders for Recommendation, TKDE 2022

---

> ### Author Rebuttal · Authors · 2023-08-08
>
> Thank you for your feedback. Below we address your concerns point by point.
>
> #### **1. On statistical efficiency and pertinence of results**
> We refer the reviewer to the global comments.
>
> #### **2. Clarification of the task-module routing matrix $Z$.**
> The task-module routing matrix has shape `(n_tasks, n_modules)`. Therefore, the information specific to task $t$ resides in the $t$-th row of this matrix. This row has shape `(n_modules,)` and selects the active modules for the $t$-th task. In the case of multiple routing heads, each head has its own routing matrix. Therefore, a given MHR adapter layer has `n_heads` $Z$ matrices, each of size `(n_tasks, n_modules)`. For a given task, we retrieve the appropriate row of each $Z$ matrix, and use the prescribed mixing weights to mix the shared modules in a task-specific way.
>
> #### **3. Improving Figure 1.**
> Thank you for the suggestions on our main figure. Let us try to better describe it. In Fig. 1, we are showing MHR with `n_heads=2`. Each head has its own task-module routing matrix $Z$. We assume in Fig. 1 that the input belongs to the first task, therefore highlight the first row of $Z$ as the active task. This row shows that the first and third module are selected, thus we select and average those modules for the first head. A similar process is down for the second head. Lastly, the outputs from the two heads are concatenated to form a LoRA adapter.
> For MHR-Z, we show an example of transfer to a new downstream task. For each head, the task-module allocation matrix (vector) has shape (`n_tasks == 1 x n_modules`). In this example, only this $Z$ vector is trained (red) while the underlying modules are frozen (blue).
>
> To improve the figure, we modified the figure to better highlight this process for the second head, and increased visibility to highlight the task specific row of the Z matrices. This version can be found in the updated pdf.
>
> Please let us know if you have further concerns or suggestions on how to improve the figure.
>
> #### **4. Clarification on Figure 3, performance of 32 heads vs 8 heads.**
> Actually, both methods perform very similarly (68.73 vs 68.79).
>
> #### **5. Addressing limitations**
> Thank you for pointing out relevant papers. Indeed, the proposed routing method can potentially benefit from leveraging additional input information (rather than task-only information). We agree that this is a promising direction for future work.

---

> > ### Comment · Reviewer_HX6c · 2023-08-19
> >
> > Thank you for your detailed rebuttal. Most of my concerns are addressed, and I would keep my rating.

---

### Author Rebuttal · Authors · 2023-08-08

Dear Reviewers, we thank you for your valuable feedback! We really appreciate the time you spent providing us constructive advice to improve our submission. We list below our general response to all of you, and address individual concerns in separate threads.


#### **1. Novelty w.r.t to prior work**

*i)* MHR is an application of multi-head splitting for routing-based approaches like Polytropon. Although multi-head splitting has been used in other contexts (notably in self-attention), its application for combining adapters is, to our knowledge, novel. Moreover, we show that it unlocks the ability to trade-off parameter efficiency and performance. This is not something that Polytropon enables by itself, as Poly-Z severely underperforms given that it relies on linear combinations of adapters.

*ii)* We also provide valuable insights on the role of routing in cross-task setups. We showcase:

*a)* **when** routing is critical. Indeed, routing is critical to multi-task optimization, but not few-shot adaptation. In fact, we can average modules, which makes MHR-$\mu$ a better initialization for LoRA than standard multi-task training. We show that MHR-$\mu$ also outperforms AdapterSoup, which relies on fixed routing. This is a new finding, and no prior work has shown that this is possible with, e.g. MoEs.

*b)* **why** routing-based methods such as Poly and MHR work, by showing that routing yields stronger gradient alignment during multi-task optimization. This finding opens the door to the design of new routing methods leveraging prior work on gradient alignment in multi-task learning [1-2].


#### **2. Statistical significance of reported improvements**

We employed a statistical analysis to assess the presence of a significant difference in performance between MHR/MHR-z and Poly. Specifically, we executed a matched-pairs Wilcoxon signed rank (WS) test. Our assessment protocol, as outlined in our primary findings (Figure 2, T5-XL backbone), was extended to encompass five separate test seeds. The outcomes distinctly reveal that MHR/MHR-z exhibit a statistical improvement (p < .05) in relation to Poly, as evidenced by the WS-test results.


[1] Yu, Tianhe, et al. "Gradient surgery for multi-task learning." Advances in Neural Information Processing Systems 33 (2020): 5824-5836.

[2] Wang, Zirui, et al. "Gradient vaccine: Investigating and improving multi-task optimization in massively multilingual models." International Conference on Learning Representations (2021)

---

### Comment · Area_Chair_PF7k · 2023-08-13
**Please respond to author rebuttals.**

Dear Reviewers,

Please respond to authors after carefully reading the author rebuttals and other reviews. If your assessment of the paper changes, please update your score with a short justification for the new rating.

The paper received diverging initial reviews. Please consider discussing with the authors or other reviewers to see whether the reviewers can reach a consensus.

Thank you,
AC

---

### Decision · Program_Chairs · 2023-09-21

**Decision:**

Accept (poster)

**Comment:**

The paper proposes Multi Head Routing (MHR) that combines subsets of parameters for a previous method called Poly, for parameter-efficient fine-tuning. The method achieves better performance with comparable number of parameters. Reviewer ratings for this paper are one accept, one weak accept and two borderline accepts. Reviewers mentioned the method is well-motivated, the empirical evidence is clear, and the presentation is high quality. The AC is pleased to recommend acceptance of the paper.